# Mass extinctions drove increased global faunal cosmopolitanism on the supercontinent Pangaea

David J. Button[1,4,5], Graeme T. Lloyd [2], Martín D. Ezcurra[1,3] & Richard J. Butler[1]

Mass extinctions have profoundly impacted the evolution of life through not only reducing taxonomic diversity but also reshaping ecosystems and biogeographic patterns. In particular, they are considered to have driven increased biogeographic cosmopolitanism, but quantitative tests of this hypothesis are rare and have not explicitly incorporated information on evolutionary relationships. Here we quantify faunal cosmopolitanism using a phylogenetic network approach for 891 terrestrial vertebrate species spanning the late Permian through Early Jurassic. This key interval witnessed the Permian–Triassic and Triassic–Jurassic mass extinctions, the onset of fragmentation of the supercontinent Pangaea, and the origins of dinosaurs and many modern vertebrate groups. Our results recover significant increases in global faunal cosmopolitanism following both mass extinctions, driven mainly by new, widespread taxa, leading to homogenous 'disaster faunas'. Cosmopolitanism subsequently declines in post-recovery communities. These shared patterns in both biotic crises suggest that mass extinctions have predictable influences on animal distribution and may shed light on biodiversity loss in extant ecosystems.

[1] School of Geography, Earth and Environmental Sciences, University of Birmingham, Edgbaston, Birmingham B15 2TT, UK. [2] School of Earth and Environment, Maths/Earth and Environment Building, The University of Leeds, Leeds LS2 9JT, UK. [3] Sección Paleontología de Vertebrados, CONICET−Museo Argentino de Ciencias Naturales "Bernardino Rivadavia", Avenida Ángel Gallardo 470, Buenos Aires C1405DJR, Argentina. [4] Present address: North Carolina Museum of Natural Sciences, Raleigh, NC 27607, USA. [5] Present address: Department of Biological Sciences, North Carolina State University, 3510 Thomas Hall, Campus Box 7614, Raleigh, NC 27695, USA. Correspondence and requests for materials should be addressed to D.J.B. (email: david.button44@gmail.com) or to R.J.B. (email: r.butler.1@bham.ac.uk)

Earth history has been punctuated by mass extinction events[1], biotic crises that fundamentally alter both biodiversity and biogeographic patterns[1, 2]. A common generalisation is that mass extinctions are followed by periods of increased faunal cosmopolitanism[1–4]. For example, the Early Triassic aftermath of the Permian–Triassic mass extinction, the largest extinction event known[5, 6], has been considered as characterized by a globally homogeneous 'disaster fauna' dominated by a small number of widely distributed and abundant taxa[1, 3, 6–8]. Similar patterns have been proposed for the aftermath of the mass extinction at the end of the Triassic[9]. However, explicit quantitative tests of changes in cosmopolitan-ism across mass extinctions are rare and have been limited to small geographical regions[3] or have not incorporated information from evolutionary relationships (phylogeny)[2, 3].

In order to test the impact of mass extinctions on biogeo-graphic patterns, a method for quantifying relative changes in cosmopolitanism through time is required. Sidor et al.[3] proposed that the spatial occurrence data can be modelled as a bipartite taxon-locality network, specifying the distribution of fossil taxa (e.g., species) within defined localities (e.g., geographic areas such as continents or basins). The biogeographic structure of this network can then be quantified. Faunal heterogeneity (or biogeographic connectedness, BC) can be measured as the rescaled density of the network—the number of taxa actually shared between localities relative to the total possible number of taxon links between them[3] (Fig. 1a, b). Higher values of BC equate to increased cosmopolitanism (i.e., less heterogeneity), whereas decreases in BC indicate increasing faunal endemism or provinciality (i.e., greater heterogeneity). This approach has been previously applied to assess regional changes in cosmopolitanism within southern Gondwana across the Permian–Triassic mass extinction[3]. Results indicated a decline in BC from the late Permian to the Middle Triassic, indicating that cosmopolitanism increased following the extinction event. However, this study did not include the critical immediate post-extinction faunas (earliest Triassic), and it is also unclear whether this regional signal is representative of global biogeographic trends.

This network method uses only the binary presence–absence data—i.e., information on whether a given species was present (and sampled) within a given locality or not. It does not explicitly incorporate information on the supra-specific phylogenetic relationships between taxa, such as could be used to estimate phylogenetic distance present between different species present at different localities. As such, it may be difficult or impossible to apply to a global fossil record dominated by singletons (species occurring at just one locality), as is common for tetrapods. Moreover, the results are potentially sensitive to systematic variation in taxonomic practice (i.e., 'lumping' vs. 'splitting') and differential temporal and spatial sampling. Consequently, it may be useful to consider how closely related sets of species from pairs of localities are on a continuous scale.

Here we present a modification of this network model that addresses these issues by incorporating phylogenetic information into the calculation of BC. Rather than treating links between taxa in different geographic regions in a binary fashion, they are instead inversely weighted in proportion to the phylogenetic distance between them (Fig. 1a, c). These reweighted links are then used to calculate phylogenetic biogeographic connectedness (pBC). As with BC, higher levels of pBC equate to more cosmopolitan faunas, with less phylogenetic distance between sets of species from pairs of localities. By contrast, lower values of pBC indicate greater endemism, and increased phylogenetic disparity between sets of species from pairs of localities. This method was applied using an informal supertree (Fig. 2a; Supplementary Note 1) and species-level occurrence data set of terrestrial amniotes ranging from the late Permian to late Early Jurassic (c. 255–175 Ma; see Supplementary Note 2). A k-means cluster analysis was used to group taxa into ten distinct geographical regions based on their occurrence palaeocoordinates (Fig. 2b; Supplementary Note 3). The sampled interval includes the Permian–Triassic and Triassic–Jurassic mass extinction events, and the origins of key terrestrial vertebrate clades such as crocodylomorphs, dinosaurs, lepidosaurs, mammaliaforms, pterosaurs, and turtles[9]. It is of particular biogeographic interest due to the presence of the supercontinent Pangaea[10], which began to break apart by the Early Jurassic. Although barriers to dispersal might be perceived as sparse on a supercontinent, numerous studies have suggested faunal provinciality and endemism on Pangaea, perhaps driven by climatic variation[3, 9, 11–13].

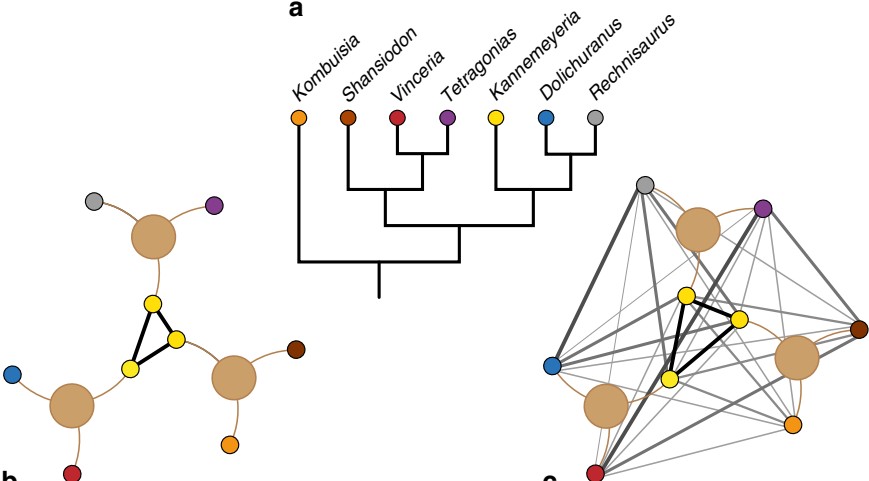

**Fig. 1** Schematic illustration of network biogeography methods. **a** Simplified phylogeny of Dicynodontia. **b**, **c** Taxon-locality networks. Localities are indicated by the large, pale brown circles, taxa are coloured as in (**a**). Taxa are connected by *brown lines* to the locality at which they occur. **b** Rescaled non-phylogenetic biogeographic connectedness (BC) of Sidor et al.[3]. A single taxon, *Kannemeyeria* (*yellow*), is present at all three localities, resulting in a link of value = 1 (*solid black line*) between each locality. **c** Phylogenetic biogeographic connectedness (pBC), as proposed here. Links (*grey lines*) between taxa from different localities are weighted inversely to their phylogenetic relatedness. *Line thickness* and *shade* is proportional to the strength of the link (and thus inversely proportional to phylogenetic distance between the two taxa)

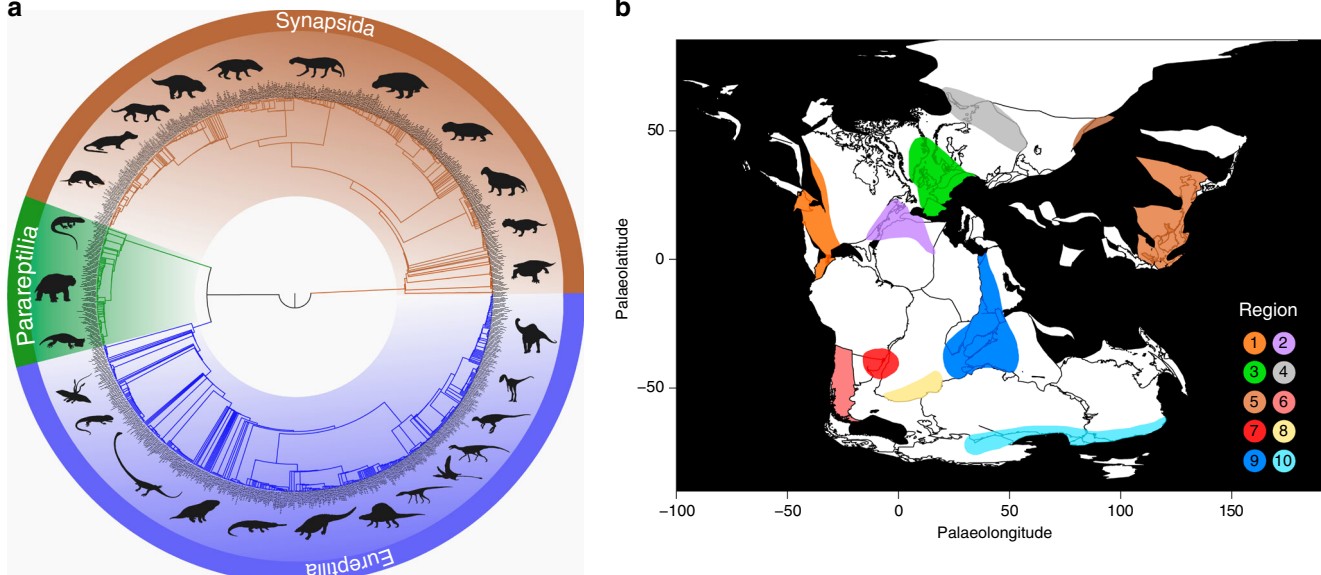

**Fig. 2** Phylogenetic framework and biogeographic regions employed in this study. **a** Informal supertree of amniotes used in the analyses. **b** Triassic palaeogeography, drawn using the 'paleoMap' R package[63] with additional reference to[10, 30], with the geographic regions used as localities for the network analysis indicated as follows. (1) Western USA, British Columbia, Mexico, Venezuela; (2) Eastern USA, Eastern Canada, Morocco, Algeria; (3) Europe, Greenland; (4) Russia; (5) China, Thailand, Kyrgyzstan; (6) Argentina; (7) Brazil, Uruguay, Namibia; (8) South Africa, Lesotho, Zimbabwe; (9) Tanzania, Zambia, Madagascar, India, Israel, Saudi Arabia; (10) Antarctica, southeast Australia

Our methodological approach allows patterns of global provincialism to be quantified, and the impact of mass extinctions on faunal cosmopolitanism tested, within an explicit phylogenetic context. The results demonstrate the evolution of relatively cosmopolitan 'disaster faunas' following both the Permian–Triassic and Triassic–Jurassic mass extinctions, suggesting that mass extinctions may have common biogeographical consequences.

## Results

**Global phylogenetic network biogeography results**. A marked and significant increase in global pBC is observed across the Permian–Triassic mass extinction (Fig. 3). A gentle, non-significant, decrease occurs from the Early Triassic to the Middle Triassic. This is followed by a strong, significant decrease to minimum pBC values (and so maximum provincialism) in the Late Triassic. A significant increase in pBC is then observed after the Triassic–Jurassic mass extinction, in the early Early Jurassic, although pBC does not reach the levels seen in the Early Triassic. pBC declines to levels similar to those seen in the Late Triassic by the end of the Early Jurassic. These results show no correlation with the number of taxa or regions sampled in each time bin (Supplementary Note 4, Supplementary Figs 1–3) and appear robust to variance in time bin length (Supplementary Figs 3d and 4).

Results for non-phylogenetic network biogeographic connectedness (non-phylogenetic BC) of the global data set significantly differ from the phylogenetic results (Fig. 3). An overall decline in non-phylogenetic BC is still observed through the Triassic, but differences between the Lopingian, Early Triassic, and Middle Triassic time bins are not significant. In addition, no increase in non-phylogenetic BC is observed over the Triassic–Jurassic boundary.

**Global analysis of taxon subsets**. An increase in global pBC across a mass extinction boundary may result from preferential survivorship of cosmopolitan lineages[8, 14–17], radiation of

opportunistic 'disaster taxa'[6], or both. In order to test which of these processes drove observed increases in global pBC, we carried out additional analyses on subsets of our data. The first set of comparisons was restricted to those less inclusive clades that exhibit high levels of survivorship across each extinction event, thereby removing the influence of preferential extinction and focusing on patterns for clades established prior to the extinction. Among these taxa, a significant change in pBC is no longer observed across the Permian–Triassic boundary (Fig. 4a), although the increase across the Triassic–Jurassic mass extinction remains significant (Fig. 4b). The second set of comparisons focused on novel, recently-diverging clades, and demonstrates very high levels of pBC for these taxa in both the Early Triassic and the earliest Jurassic, significantly greater than total pBC in both these and the preceding time bins (Fig. 4a, b). Comparison of recently diverging clades in all time bins recovers the same signal as that from the total data set (Supplementary Note 5, Supplementary Fig. 5), indicating that variation in pBC is not a result of differences in average clade age in each time bin.

**Geographically localized analyses**. To compare hemispherical trends in biogeographic connectedness, pBC was also calculated for Laurasia and Gondwana separately. The signal from Laurasian occurrences matches very closely with the global pattern (Fig. 5a). By contrast, patterns in Gondwana diverge markedly from global trends in the latest Triassic, where pBC abruptly rises, and then gradually declines through the Early Jurassic (Fig. 5a).

In addition, pBC analysis was implemented on terrestrial amniote occurrences from the southern Gondwanan data set of Sidor et al.[3]. This data set groups taxa at a geological basin, rather than broader regional, level; as a consequence, this analysis indicates how pBC differs at geographically smaller scales. Biogeographic connectedness is lower in the Middle Triassic than in the late Permian under both phylogenetic and non-phylogenetic treatments of these data (Fig. 5b); however, the result is not significant for phylogenetic BC.

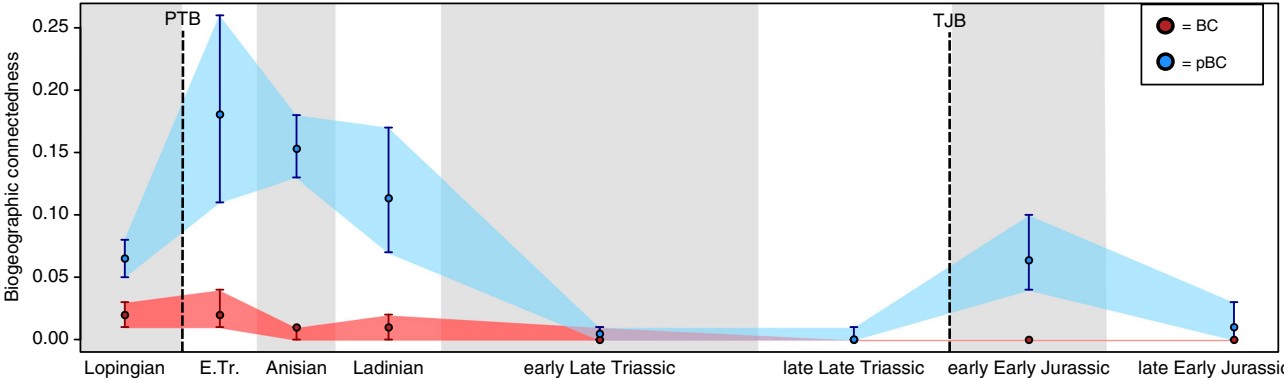

**Fig. 3** Results from BC analysis of Lopingian-Early Jurassic terrestrial amniotes. Results from both non-phylogenetic (BC, *red*) and phylogenetic (pBC, *blue*) analyses of global biogeographic connectedness are shown. *Shaded* polygons represent 95% confidence intervals (calculated from jackknifing with 10,000 replicates) for both the BC and pBC analyses. The Permian–Triassic boundary (PTB) and Triassic–Jurassic boundary (TJB) extinction events are indicated by *dotted lines*. E. Tr.: Early Triassic

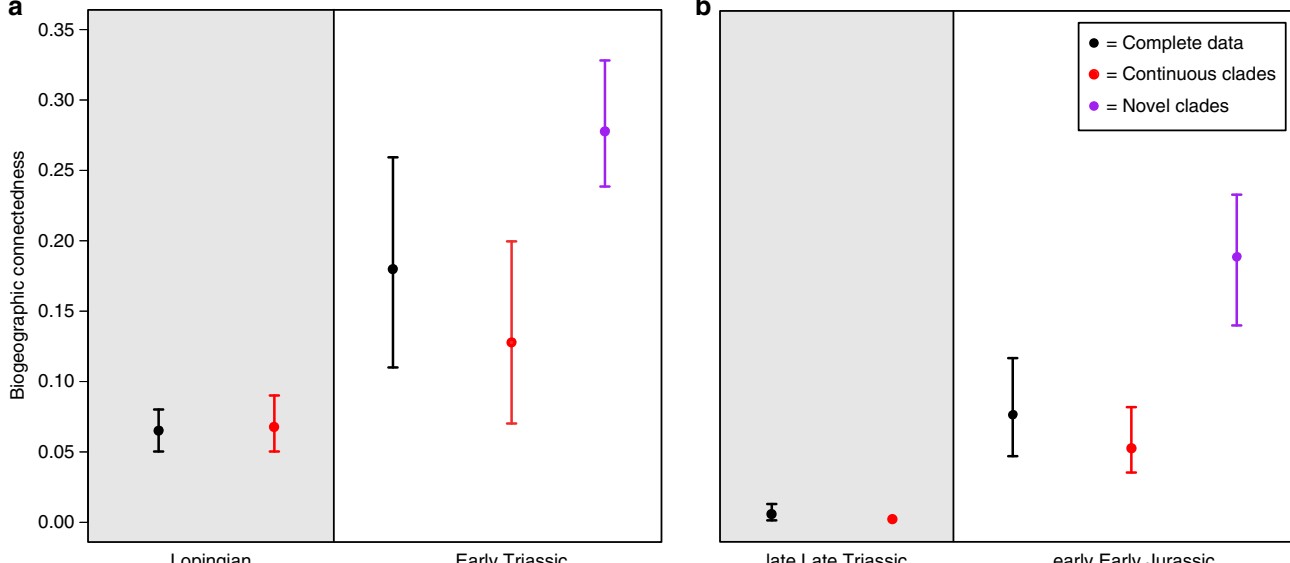

**Fig. 4** Results from BC analysis of taxonomic subsets. Comparison of results for the data subsets across the Permian–Triassic **a** and Triassic–Jurassic **b** mass extinctions. Results for the entire data set are in *black*, those for less inclusive clades showing high survivorship in *red*, and those for the most recently diverging taxa in *purple*. Ninety-five percent confidence intervals, calculated from jackknifing with 10,000 replicates, are indicated

## Discussion

The Triassic represents an important time in the evolution of vertebrate life on land. It witnessed a series of turnover events that resulted in a major faunal transition from Palaeozoic communities, dominated by non-mammalian synapsids and parareptiles, to more modern faunas, including clades such as crocodylomorphs, dinosaurs, lepidosaurs, mammaliaforms, and turtles[9, 18]. Our novel phylogenetic network approach helps to place these major faunal transitions of the Triassic within a global biogeographical context by allowing changes in faunal connectivity to be quantified within an explicit evolutionary framework.

Our results demonstrate an overall decrease in pBC from the Lopingian to the Early Jurassic, but punctuated by significant increases across both the Permian–Triassic and Triassic–Jurassic mass extinction events. This provides quantitative support for classically held hypotheses about the presence of a global cosmopolitan fauna in the aftermath of and in response to these events[2, 3]. The robustness of these results to sampling variation

and variable time bin length supports their interpretation as real biogeographical signals.

Our taxon subset analyses were explicitly aimed at disentangling the impact of alternative mechanisms that could lead to this pattern of increased post-extinction pBC. Novel clades, those diverging immediately prior to or immediately after each mass extinction, were analysed separately and exhibit relatively high levels of pBC (i.e., increased cosmopolitanism relative to the preceding time bin) in both the Early Triassic and earliest Jurassic (Fig. 4a, b). By contrast, surviving clades, those well-established prior to the extinction and extending through it, exhibit no increase across the Permian–Triassic boundary and only a moderate increase across the Triassic–Jurassic boundary (Fig. 4b). This indicates that the increases in pBC following each extinction were primarily driven by the opportunistic radiation of novel taxa to generate cosmopolitan 'disaster faunas', rather than being due to preferential extinction of endemic taxa[19]. Recently-diverging clades in other time bins do not exhibit elevated pBC (Supplementary Note 5) and there is no correlation

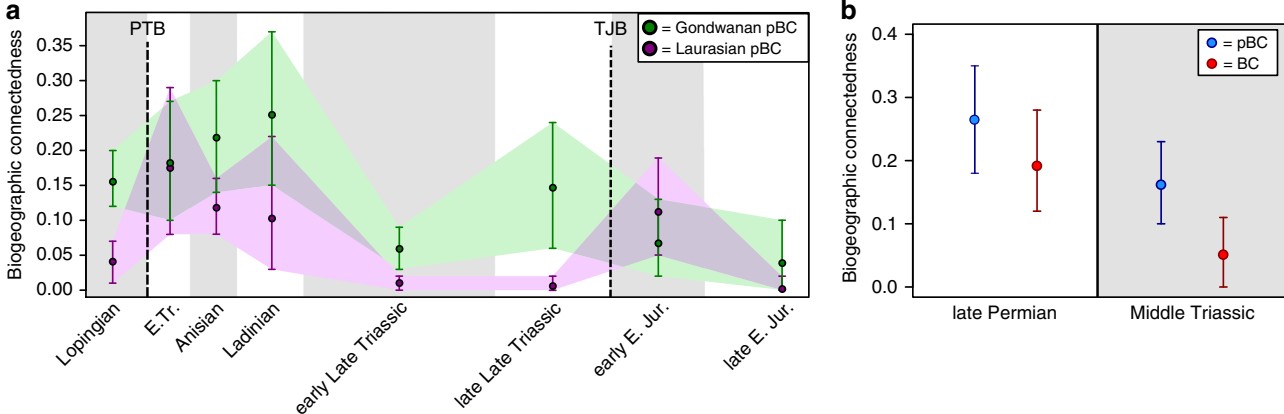

**Fig. 5** Results from BC analysis of geographically localized areas. **a** Comparison of pBC trends during the Lopingian-Early Jurassic from Gondwanan localities (in *green*) against those for Laurasia (in *purple*). Ninety-five percent confidence intervals, calculated from jackknifing with 10,000 replicates, are indicated. **b** Results from analysis of basin-level terrestrial amniote occurrences from the late Permian and Middle Triassic of southern Pangaea, from the data set of Sidor et al.[3]. Phylogenetic BC results are given in *blue*, non-phylogenetic BC in *red*. Ninety-five percent confidence intervals, calculated from jackknifing with 1000 replicates, are indicated. Abbreviations as in Fig. 3; E. Jur.: Early Jurassic

between pBC and average branch length in each time bin (Supplementary Note 6, Supplementary Fig. 6), indicating that this result is due to abnormal conditions following each mass extinction as opposed to being a property of clade age.

The global biogeographic restructuring of biological communities associated with these mass extinction events hence provides evidence of the release of biotic constraints[3], which would have facilitated the radiation of new or previously marginal groups, such as archosauromorphs following the Permian–Triassic mass extinction[3], and dinosaurs and mammaliaforms during the Early Jurassic[20, 21]. This highlights the importance of historical contingency in the history of life, where unique events such as mass extinctions have exerted strong influences on the subsequent macroevolutionary patterns observed in deep time[22–24].

The global pBC pattern recovered here differs from the more geographically focused and temporally limited non-phylogenetic study of Sidor et al.[3], which found Middle Triassic levels of BC in southern Pangaea to be lower than those seen in the late Permian. Reanalysis of the amniote occurrences from the basin-level data set of Sidor et al.[3] demonstrates that pBC also declines between these time bins, although not significantly (Fig. 5b). Looking more broadly, pBC trends in Gondwana differ from those seen in Laurasia (Fig. 5a). This is particularly evident in the Late Triassic and Early Jurassic, in which a significant increase and decrease in pBC is seen in Laurasia for each time bin, respectively, but not in Gondwana (Fig. 5a).

These results suggest that localized biogeographic patterns within Gondwana may have been decoupled from those seen elsewhere in the northern hemisphere. This would corroborate previous work, suggesting the evolution of a distinct fauna, that includes massopodan sauropodomorphs, ornithischians, basal saurischians, and prozostrodontian cynodonts as relatively common taxa in South America and Africa during the Late Triassic[11]. The occurrences of guaibasaurids[25] and floral similarities[26, 27] provide some links between South American communities and the upper Maleri Formation of India, although the latter assemblage remains relatively poorly-known and sampled. The Triassic–Jurassic mass extinction was a global event[19] and it is unclear why decoupling of biogeographic trends within Gondwana should occur. Sampling within Gondwana during this interval is uneven, with the bulk of occurrences coming from palaeolatitudes between 30–60°S (see Supplementary Note 4). During the Late Triassic the 30–60° latitudinal belts

were dominated by subtropical desert[28]. Interestingly, whereas this biome was more fragmented by seasonally wet conditions through into the Jurassic within Laurasia, it remained relatively stable in Gondwana[26, 28]. It is possible that this stability may have contributed to the evolution of a distinct fauna in the southern hemisphere. Alternatively, however, this distinct Gondwanan pattern may be a sampling artefact. Although the inclusion of phylogenetic information allows the approach used here to incorporate more data than previous methods, sampling of latest Triassic and earliest Jurassic Gondwanan localities is relatively poor and uneven, leading to the low statistical power of results within these time bins. In the earliest Jurassic, in particular, over 80% of Gondwanan tetrapod occurrences are from the upper Elliot and Clarens formations of South Africa. Further evaluation of this possible signal will require sampling of new Late Triassic and Early Jurassic Gondwanan localities, particularly from India and Antarctica.

Under our non-phylogenetic network analysis of the global data set, no increase in BC is observed across the Triassic–Jurassic boundary; indeed, no significant differences are observed between any consecutive time bins (Fig. 3). This highlights the importance of including phylogenetic information in global analyses such as that conducted here; without the incorporation of phylogeny, aspects of biogeographic signal may be obscured. The decline of pBC to minimal levels towards the end of the Triassic supports hypotheses of strong faunal provinciality and increased endemism within Pangaea during the early Mesozoic[3, 9, 12, 13, 29]. The distribution of Late Triassic tetrapods varies with latitude[9, 11–13], a pattern also observed in terrestrial floras[9, 27]. This is somewhat unexpected, given that oceanic barriers to dispersal were scant[30] and the latitudinal temperature gradient was weak[28] in Pangaea during the Late Triassic. Instead, the 'mega-monsoonal' climate of Late Triassic Pangaea[28] would have driven provinciality of faunas through strong latitudinal and seasonal variation in precipitation[12, 13]. Patterns of endemism farther back into the Palaeozoic are presently unclear because the Lopingian was preceded by a poorly-understood period of taxonomic turnover during the Guadalupian[31]. Analysis of older Palaeozoic time bins will be required to elucidate changes in endemism during the earlier history of Pangaea.

This background trend of increasing endemism contrasts sharply with the increase in pBC immediately following each mass extinction. This highlights the unique macroevolutionary

regimes associated with mass extinctions[24, 32], with post-extinction 'disaster faunas' being the result of the abnormal selective conditions operating in the wake of these crises. An increase in global cosmopolitanism, with a prevalence of 'disaster taxa', has also been observed in marine invertebrates across the Ordovician-Silurian[33, 34], Permian–Triassic[35, 36], and Cretaceous-Palaeogene[14] mass extinctions, although these studies have not explicitly incorporated phylogenetic data. Similarly, more generalized insect-plant associations show higher survivorship across the Cretaceous-Tertiary mass extinction[37] and, on the smaller scale, Pleistocene-Holocene warming resulted in a greater unevenness of small mammal faunas in northern California[38]. Our demonstration of a similar signal in terrestrial communities in the latest Palaeozoic and early Mesozoic suggests that mass extinctions exert predictable biogeographical influences. However, the Permian–Triassic and Triassic–Jurassic events may be unique amongst terrestrial mass extinctions due to the presence of Pangaea, where the perceived reduction in barriers to overland dispersal might have facilitated the development of high levels of terrestrial cosmopolitanism. Extending the methodology employed here to other extinction events, such as for terrestrial faunas across the Cretaceous–Palaeogene boundary, will provide further tests of generalizable biogeographic trends across different mass extinction events.

These common trends observed in the fossil record have the potential to inform modern conservation efforts, given that the current biodiversity crisis is acknowledged as representing another mass extinction event[39]. Global homogenisation due to human activities, such as landscape simplification[40], ecosystem disruption[40–42], anthropogenic climate change[4, 38, 42], and introduction of exotic species[42–44], represents a principal threat to contemporary biodiversity[43, 45]. Ongoing extinction will exacerbate this[42, 43] with a shift towards a more generalized 'disaster' fauna projected on the basis of current trends[4, 46]. The observation of global collapse in biogeographic structure accompanying previous mass extinctions, as documented here, corroborates this and is of key importance in forecasting the biological repercussions of the current biodiversity crisis.

## Methods

**Phylogeny.** An informal supertree of 1046 early amniote species ranging from 315–170 Ma was constructed from pre-existing phylogenies (Fig. 2a; Supplementary Note 1, Supplementary Data 1). We used an informal supertree approach rather than a formal supertree in order to maximize taxonomic sampling, including species that have not been included in quantitative phylogenetic analyses. In addition to the taxa included in the biogeographic connectedness analyses, this sample included some stratigraphically older taxa in order to more accurately date deeper nodes. In order to account for phylogenetic uncertainty, 100 time-calibrated trees, with random resolution of polytomies, were produced from this supertree utilizing the 'timePaleoPhy' function of the paleotree package[47] in R (version 3.2.3;[48]). Trees were dated according to first occurrence dates, with a minimum branch length of 1 Myr.

**Taxon occurrences and ages.** A global occurrence database of 891 terrestrial amniote species was assembled, primarily from the Paleobiology Database[49], with the addition of some occurrences from the literature (see Supplementary Note 2, Supplementary Data 2). Taxa were dated at stage level. They were then placed in the following time bins for analysis: Lopingian, Early Triassic (Induan and Olenekian), Anisian, Ladinian, early Late Triassic (Carnian–early Norian), late Late Triassic (late Norian–Rhaetian), early Early Jurassic (Hettangian, Sinemurian), and late Early Jurassic (Pliensbachian, Toarcian). The Late Triassic was not split into its constituent stages due to the disproportionately long Norian stage:[50–53] rock units from this epoch were instead assigned to either the early Norian or the late Norian (Supplementary Tables 1, 2).

**Geographic areas.** In order to conduct network and many other palaeobiogeographic analyses, it is necessary to identify a series of geographically discrete areas (the localities of the taxon-locality network in the network methodology). These areas are typically defined solely on the basis of geography (rather than shared flora or fauna) because the aim is to test faunal similarity between geographically distinct regions of the world. For example, previous analyses have commonly used modern

continents as input areas[10, 11, 13, 15]. This traditional approach is potentially problematic on a supercontinent where, for example, eastern North American and north-western African localities were much closer to each other than to localities in southwestern North America or southern Africa. Instead, we defined our geographic areas on the basis of k-means clustering of the palaeocoordinate data for 2144 terrestrial fossil occurrences from the relevant time span, obtained mostly from the Paleobiology Database (Supplementary Note 3). Importantly, this approach does not require or use any information on taxonomy or phylogeny—it is solely designed to find geographically-discrete clusters of fossil localities—and thus, it is fully independent from the subsequent network analyses.

The data were binned at epoch level, with each epoch analysed separately to avoid confusion arising from continental movements. K-means clustering was performed within R, varying the value of $k$ from 5–15. For each value of $k$, the analysis was repeated with ten random starts, with 100 replicates). Performance of different analyses was then compared on the basis of the percentage of variance explained, and results were compared with palaeogeographic reconstructions through this interval[10, 54] (Supplementary Table 3; full results are given as Supplementary Data 3). This resulted in the designation of ten discrete palaeogeographic regions that each represent localities for the network analyses (Fig. 1b). Taxa were assigned to one or more regions as appropriate, yielding a taxon-locality matrix for each time bin (Supplementary Data 4).

**Phylogenetic network biogeography analyses.** Non-phylogenetic biogeographic connectedness(BC) was previously quantified[3] as the rescaled density of a taxon-locality matrix, calculated as follows:

$$BC = \frac{O - N}{(L*N) - N}. \tag{1}$$

In this formula, $O$ = the number of links in the network (the sum of all values in a taxon-locality matrix, which will equal the number of occurrences in a non-phylogenetic analysis), $N$ = the number of taxa, and $L$ = the number of localities. This gives the ratio between the number of taxa present beyond a single locality and the maximum possible number of occurrences (i.e., every taxon present at every locality). Aside from whether a taxon is identical or not, no further phylogenetic information is included using this method—links are only considered where an individual taxon is shared between different localities, and are all equally weighted.

Herein, this method was modified to include phylogenetic information (pBC) by weighting links between taxa as inversely proportional to the phylogenetic distances between them. Phylogenetic distances between taxa were measured by summing the branch lengths in millions of years representing the shortest distance between two taxa. This was then scaled against the maximum possible phylogenetic distance (i.e., the total distance of the summed branch lengths between the two most distantly related taxa). This scaled value was then subtracted from one to yield the weight of each link: the values of links between taxa hence vary between one (co-occurrence of the same species in two separate localities) and zero (when comparing the two most distantly related taxa in the taxon-locality matrix). The sum of the reweighted taxon-locality matrix was then substituted for $O$ in Eq. 1 to yield a value of pBC. This method has been made available as the "BC" function within the R package dispeRse[55] (available at github.com/laurasoul/dispeRse): example analysis scripts are given as Supplementary Data 5 and Supplementary Data 6. It should be noted that a given value of pBC will be a non-unique solution: the same value could theoretically be generated by many links between distantly-related taxa or by fewer links between more closely-related species. Disentangling these possibilities is difficult. However, comparison of results with measured phylogenetic distances and number of taxa in each time bin indicates that pBC results are not merely driven by differences in the relatedness of sampled taxa, and instead reflect genuine biogeographical signal (see supplementary information).

Analysis of a simulated null (stochastically generated) data set indicated a predictable and systematic pattern of increasing pBC through time. This is due to the increasing distance from a persistent root to the tips through time, resulting in phylogenetic branch lengths between nearest relative terminal taxa becoming proportionally shorter. In order to compare pBC between different time bins, it is therefore necessary to remove this tendency for pBC to increase in later time bins. We achieved this through the introduction of a constant, $\mu$, which collapses all branches below a fixed "depth" such that root age is equal to $\mu$ million years before the tips. The introduction of this constant also alleviates problems of temporal superimposition of biogeographic signals that may otherwise occur. It means that pBC results reported for each time bin reflect patterns generated by biogeographic processes in the preceding $\mu$ million years, preventing these recent biogeographic signals of interest from being swamped by those from deeper time intervals. A $\mu$ value of 15 was chosen based on the results of sensitivity analyses varying the value of $\mu$ from 5–25 Myr in 1 Myr increments (Supplementary Note 7, Supplementary Fig. 7).

This method was applied to the taxon-region matrix for each time bin, and the 100 time-calibrated supertrees, pruning taxa not present within the bin of interest (effectively making each tree ultrametric) to calculate pBC. Jackknifing, with 10,000 replicates, was used to calculate 95% confidence intervals. This analysis was then repeated without phylogenetic information to gauge the importance of phylogeny on observed patterns.

**Taxon subset analyses**. In order to investigate the processes giving rise to observed changes in cosmopolitanism over mass extinction events, analyses were also performed on two taxonomic subsets. The first reanalysed time bins either side of each mass extinction (the Lopingian and Early Triassic and late Late Triassic and early Early Jurassic), including only small clades exhibiting high survivorship (<20 species, with ≥ 20% of lineages crossing the extinction boundary). This was intended to minimize the influence of possible preferential extinction of geographically-restricted taxa.

The removal of taxa during mass extinctions opens new vacancies in ecospace, promoting adaptive radiations in surviving, often previously marginal, clades[56, 57]. For example, the Permian–Triassic mass extinction is seen as a causal factor in the succeeding radiation of epicynodonts[58] and archosaurs[3, 59, 60], and the Triassic–Jurassic radiation as pivotal in the diversification of crocodylomorph[61] and dinosaur clades[20, 62]. 'Disaster faunas' will hence be expected to be composed of relatively recently diverging clades, as surviving taxa diversify into broader geographic ranges (e.g., ref. [59]). To test the significance of this, we reanalysed the time bins immediately following each mass extinction, including only clades that branched <2 Myr prior to or after the boundary. In order to ensure that the results of this analysis reflected differences in the post-extinction bins as opposed to an artefact of clade age, also performed analyses applying this filter to the other time bins (see Supplementary Note 6).

**Geographically localized analyses**. To atomise global pBC signals into hemispheric trends, pBC was re-calculated for Laurasian and Gondwanan areas separately following an identical procedure to that for global analyses. Finally, to compare global results obtained from this new method with the more localized analysis of Sidor et al.[3], another set of analyses was performed following the taxonomic sampling of the latter. Terrestrial amniote occurrences from the late Permian and Middle Triassic of the Karoo Basin of South Africa; Luangwa Basin of Zambia; Chiweta beds of Malawi; Ruhuhu Basin of Tanzania, and the Beacon Basin of Antarctica were taken from the data set of Sidor et al.[3]. These data and the 100 time-calibrated trees described above were then used to calculate BC and pBC between these basins for each of the sampled time bins.

**Data availability**. All the data analysed in this study and example code are available in the supplementary data files.

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

## Acknowledgements

We thank R. Benson, R. Close, D. Cashmore, and E. Dunne for discussion. This research received funding from the Marie Curie Actions (grant 630123 to R.J.B.), an ERC Starting Grant (grant 637483 to R.J.B.), and a Discovery Early Career Researcher Award (grant DE140101879 to G.T.L.). This is Paleobiology Database official publication 289.

## Author contributions

G.T.L., R.J.B and M.D.E.: Conceived the research. G.T.L. and R.J.B.: Wrote new functions as required for these analyses. D.J.B.: Compiled the data, performed the analyses and prepared the figures. All authors discussed results and contributed to writing the manuscript.

## Additional information

**Competing interests:** The authors declare no competing financial interests.

