## [Peer Review File · Nature Communications]

Reviewers' comments:

Reviewer #1 (Remarks to the Author):

This manuscript quantifies changes in the biogeographic patterns of vertebrates across two mass extinction events and find support for the increase of cosmopolitanism following mass extinction events, strongly influenced by the diversification of disaster faunas. The authors used a quantitative approach to corroborate this evolutionary pattern, which had been previously identified in qualitative studies.

This manuscript has several merits that make it an innovative and original contribution. First, the authors propose a refinement of a previously proposed method that can be applied to extinct faunas for measuring their affinities but taking into account the phylogenetic relationships of the taxa that compose these faunas. Second, in contrast to most studies the authors did not assume continental-wide regionalization of taxa based on modern day land-masses but on paleogeographic proximity based on a cluster analysis. Third, they apply this study to a massive dataset of terrestrial vertebrates so that these extinction events are tested quantitatively for the first time and for a dataset of close to a thousand species.

The methodological improvements will surely spark the use of this method in other scenarios and problems and by other researchers so that the potential impact of this manuscript goes beyond the biogeographic pattern found in the Permo-Triassic and Triassic-Jurassic mass extinction events.

I have two main comments that the authors should consider:

1) The results provided by not only corroborate previously suspected patterns (i.e., increased cosmopolitanism following mass extinctions) but also show an intriguing decoupling of the pattern observed in the northern vs southern hemisphere. There has been an increasing amount of studies showing provinciality in the Triassic across Pangea, but finding that there is a biogeographic decoupling in mass extinction events is remarkable. This is to me unexpected and may have important implications because mass extinction events (especially the Permo-Triassic) are thought to be worldwide events and I am not sure why you would expect a different pattern in the northern and southern hemisphere. This issue is only slightly mentioned and its implications are not further discussed. It would be interesting to further explore this topic.

2) The authors have manually assembled "informal" supertree of 1046 taxa, which surely implied a remarkably laborious task and a large part of the supplementary is dedicated to explain what decisions have been made to place certain taxa in certain places of the tree. There are a couple of issues I find problematic with this approach. First, there is no discussion at all about why the authors manually constructed this supertree instead of using one of the various methodologies there are available for this process. Surely all the supertree methods have their own characteristics and even biases and problems, but a great deal of research has been published on developing quantitative methods to construct supertrees based on source phylogenies. Given the authors are proposing a new method, it would be desirable to provide a more complete protocol that other authors can follow. Why they didn't use a supertree method? Are they not adequate for creating supertrees that can be used as input for a pBC analysis?

If I want to apply this method for another case/problem it is not clear to me if the authors consider currently available quantitative supertree methods as problematic (and why) or if they have just preferred to do it manually for their study for other reason (and what is that reason).

A second issue with the manually constructed supertree is that it involves many decisions that are

subjective and supertree methods may define differently the uncertainty regarding the relationships of certain taxa. This may or may not affect the biogeographic pattern of mass extinctions discovered by the authors, but this point is left unexplored. It would be interesting to test how sensitive are their results and conclusions to the use of their manually constructed supertree as input for the analysis. If you use a different supertree, such as one obtained through a supertree method, what do you obtain. For instance, if you apply semi-strict supertree method, do you obtain a similar pattern in the pBC graphs? What happens if you use any other supertree method? Sometimes the phylogenetic uncertainty causes minor effects on macroevolutionary or biogeographical studies but sometimes its effect can be pervasive. It would be very good to know in which of the two categories falls the biogeographic pattern discovered in this manuscript.

Despite these two issues that I would like to see further developed, I think this paper will influence the way we approach studies on the biogeographic patterns during mass extinctions. Much of previous work on this topic has neglected the influence of phylogeny and I consider this manuscript a decisive step in the right direction. It will make future workers use methods like the one proposed here or at least use a tree-thinking approach to this relevant problem.

Reviewer #2 (Remarks to the Author):

This manuscript is a valuable contribution on an important topic: the effects of extinctions on the geographic distribution of terrestrial organisms. The incorporation of phylogenetic relatedness is a novel development and should provide more accurate interpretations relative to earlier metrics (particularly in sparse groups like terrestrial tetrapods). I do have a few queries about the methods, outlined below (which may or may not reflect my ignorance of phylogenetic methods than anything else...). I think the main conclusions are well-supported by the data and the manuscript lays out the data and interpretations clearly. A few of the wider interpretations about the causes might be a stretch; for example lines 198-199: is there evidence for abnormal selective conditions, as opposed to perhaps broader environmental tolerances of newly-evolved clades? That's somewhat of a minor issue that doesn't detract from the overall claim of the manuscript, which I think will be of broad interest.

If I understand the method accurately, it seems that average clade age may constrain the possible range of pBC. If most branching points are deep back in time, pBC is constrained to be small (in the extreme, but presumably unrealistic, case where all branching points occurred earlier than the cutoff time μ , the value of pBC would be zero - I think?). Presumably this effect explains the slopes in figure S1. If this is the case, do your results simply measure clade age rather than ecological or biogeographic processes that promote cosmopolitan distributions? I expect that many new clades should radiate in the aftermath of extinctions. I suppose that there is still some biogeographic signal, as related taxa need to be found in separate regions (rather than the clade only radiating within a single region) to increase pBC, but do you have a sense for the relative magnitude of a biogeographic signal above and beyond just having younger clades?

On a related note, if clade age is important, do recently-evolved clades consistently exhibit greater pBC? Figure 4 shows that novel clades in the Early Triassic and early Early Jurassic have greater pBC, but would that be true in all time intervals? I don't imagine that the absolute pBC value for novel clades would be as high in the Lopingian as it is in the Early Triassic, but I wonder if there would be some offset between continuous and novel clades in all time periods.

More generally, is the pBC value a non-unique solution? That is, could the same value arise from a situation with more links among moderately-related taxa and also from one with fewer links among more closely-related taxa? When thinking about a cosmopolitan disaster fauna, I don't imagine most

people would picture moderately-related taxa occurring in different regions. Would it be possible to parse the pBC value somehow to separate the contributions of taxa at different levels of relatedness? I suppose this is kind of a fuzzy question – maybe it wouldn't provide any insights. That said, the values are less intuitive than the non-phylogenetic version (it's easy to interpret when the exact taxon is shared), so it might be useful to help people gain that intuitive sense of the change in biogeographic connectedness (especially people like me who aren't as well-versed in phylogenetic thinking).

A minor point, on lines 324-326: when the term "disaster taxa" was first coined, it referred to long-ranging organisms that were typically rare or restricted but that thrived in the extinction aftermath. These were things like stromatolites and *Lingula* (or really "Lingula") after the Permian-Triassic extinction in the marine realm. The term is certainly used more broadly now, but I don't know that I would have assumed recently-diverging clades to comprise the disaster fauna. Your results may support the idea that recently-diverging clades have less biogeographic differentiation (using that definition of a disaster fauna), assuming it's not just a signal of average clade age.

Lines 150-152. At the risk of getting into semantics, is a pBC value of 0.05 (for the early Early Jurassic) really a global cosmopolitan fauna? If a value of 1 is truly cosmopolitan (albeit unrealistically so), 0.05 seems much less!

Finally, I'm not entirely convinced that there is a robust long-term trend towards more endemism. Yes, the Lopingian pBC value is higher than the Late Triassic or late Early Jurassic, but the time-series is quite short. The Lopingian is also not long after a turnover in tetrapod faunas (Olson's extinction). Without having a more data in the Paleozoic, I might be more circumspect about the significance of a background trend in endemism.

Sincerely,

Matthew Clapham

We thank both reviewers for their constructive and helpful comments. We believe that we have satisfied the suggestions made by the reviewers, strengthening both our analyses and the overall quality of the manuscript. A point-by-point response to each of the reviewer's comments is given below.

Reviewer #1 (Remarks to the Author):

"This manuscript quantifies changes in the biogeographic patterns of vertebrates across two mass extinction events and find support for the increase of cosmopolitanism following mass extinction events, strongly influenced by the diversification of disaster faunas. The authors used a quantitative approach to corroborate this evolutionary pattern, which had been previously identified in qualitative studies.

This manuscript has several merits that make it an innovative and original contribution. First, the authors propose a refinement of a previously proposed method that can be applied to extinct faunas for measuring their affinities but taking into account the phylogenetic relationships of the taxa that compose these faunas. Second, in contrast to most studies the authors did not assume continental-wide regionalization of taxa based on modern day land-masses but on paleogeographic proximity based on a cluster analysis. Third, they apply this study to a massive dataset of terrestrial vertebrates so that these extinction events are tested quantitatively for the first time and for a dataset of close to a thousand species.

The methodological improvements will surely spark the use of this method in other scenarios and problems and by other researchers so that the potential impact of this manuscript goes beyond the biogeographic pattern found in the Permo-Triassic and Triassic-Jurassic mass extinction events.

I have two main comments that the authors should consider:

1) The results provided by not only corroborate previously suspected patterns (i.e., increased cosmopolitanism following mass extinctions) but also show an intriguing decoupling of the pattern observed in the northern vs southern hemisphere. There has been an increasing amount of studies showing provinciality in the Triassic across Pangea, but finding that there is a biogeographic decoupling in mass extinction events is remarkable. This is to me unexpected and may have important implications because mass extinction events (especially the Permo-Triassic) are thought to be worldwide events and I am not sure why you would expect a different pattern in the northern and southern hemisphere. This issue is only slightly mentioned and its implications are not further discussed. It would be interesting to further explore this topic."

Our discussion of this result was brief in the original version of the manuscript because the Gondwanan results for the Late Triassic and Early Jurassic have very broad confidence intervals, complicating the interpretation of patterns – as such, we were reticent to place substantial focus on these results. Nevertheless, this finding and its possible causes have now been expanded on in lines 187-209. We discuss some possible biological explanations for the pattern, but also the possibility that the pattern may result from incomplete sampling.

"2) The authors have manually assembled "informal" supertree of 1046 taxa, which surely implied a remarkably laborious task and a large part of the supplementary is dedicated to explain what decisions have been made to place certain taxa in certain places of the tree. There are a couple of issues I find problematic with this approach. First, there is no discussion at all about why the authors manually constructed this supertree instead of using one of the various methodologies there are available for this process. Surely all the supertree methods have their own characteristics and even biases and problems, but a great deal of research has been published on developing quantitative methods to construct supertrees based on source phylogenies. Given the authors are proposing a new method, it would be desirable to provide a more complete protocol that other authors can follow. Why they didn't use a supertree method? Are they not adequate for creating supertrees that can be used as input for a pBC analysis?

If I want to apply this method for another case/problem it is not clear to me if the authors consider currently available quantitative supertree methods as problematic (and why) or if they have just preferred to do it manually for their study for other reason (and what is that reason).

A second issue with the manually constructed supertree is that it involves many decisions that are subjective and supertree methods may define differently the uncertainty regarding the relationships of certain taxa. This may or may not affect the biogeographic pattern of mass extinctions discovered by the authors, but this point is left unexplored. It would be interesting to test how sensitive are their results and conclusions to the use of their manually constructed supertree as input for the analysis.

If you use a different supertree, such as one obtained through a supertree method, what do you obtain. For instance, if you apply semi-strict supertree method, do you obtain a similar pattern in the pBC graphs? What happen if you use any other supertree method? Sometimes the phylogenetic uncertainty causes minor effects on macroevolutionary or biogeographical studies but sometimes its effect can be pervasive. It would be very good to know in which of the two categories falls the biogeographic pattern discovered in this manuscript.

Despite these two issues that I would like to see further developed, I think this paper will influence the way we approach studies on the biogeographic patterns during mass extinctions. Much of previous work on this topic has neglected the influence of phylogeny and I consider this manuscript a decisive step in the right direction. It will make future workers use methods like the one proposed here or at least use a tree-thinking approach to this relevant problem."

We should begin by noting that our phylogenetic hypothesis is a supertree – it was constructed by synthesising other, smaller phylogenetic hypotheses – but note that strictly speaking it is an informal supertree as it was generated "manually", rather than through an algorithm (the formal supertree approach). Thus it is not novel and indeed has been used in several other publications [e.g. Benson & Choiniere, 2013; Huttenlocker, 2014; Puttick *et al.*, 2014; Ezcurra *et al.*, 2016; Foth & Joyce, 2016; Foth *et al.*, 2016; Stubbs & Benton, 2016; Button *et al.*, 2017]. Although it is outside the scope of this publication we would note that there are strengths and weaknesses to both approaches (for a discussion see particularly the supplementary information for Lloyd *et al.* 2016). Briefly though, a formal supertree would allow us to more appropriately quantify phylogenetic uncertainty and (usually, but see below) maximise taxonomic inclusion. However, appropriate protocols (such as that of Lloyd *et al.* 2016, see above) are labour intensive and would necessitate far more work than the reviewer implies. We would also note that it is the aim of one of the authors (GTL) to produce such a hypothesis in future and we refer the reviewer to a data set being assembled for this purpose on his web site (graemetlloyd.com/matr.html). Nevertheless, there are reasons to prefer the informal supertree used here. First, it allowed us to maximise data inclusion as it allowed us to incorporate a large number of taxa and taxonomic occurrences that have not previously been included in quantitative phylogenetic analyses – but their phylogenetic relationships are based on non-quantitative interpretations – and so could not be included in a formal supertree approach. It also allowed us to maximise phylogenetic resolution within clades such as archosaurs and synapsids where there has been historically substantial phylogenetic conflict but which have recently received significant anatomical and phylogenetic appraisal (e.g., Nesbitt, 2011; Kammerer *et al.*, 2013; Ezcurra, 2016). We have added some text into the main manuscript to

explain this justification. It should also be noted that our analyses did include sensitivity testing of phylogenetic uncertainty via the generation of 100 trees in which polytomies were randomly resolved.

Importantly, the downstream effects of our decision to use an informal rather than formal supertree do not invalidate our overall modelling approach, as this approach can be applied to any phylogenetic hypothesis. Nor would it, we strongly suspect, fundamentally change our results or conclusions. Beyond that, our main aim in this manuscript is to present a new biogeographic network approach and test macroevolutionary hypotheses related to mass extinctions, but not to compare the results from informal versus formal approaches. Nevertheless, this would be an interesting avenue for future work and something one of us (GTL) may pursue (see above).

Reviewer #2 (Remarks to the Author):

“This manuscript is a valuable contribution on an important topic: the effects of extinctions on the geographic distribution of terrestrial organisms. The incorporation of phylogenetic relatedness is a novel development and should provide more accurate interpretations relative to earlier metrics (particularly in sparse groups like terrestrial tetrapods). I do have a few queries about the methods, outlined below (which may or may not reflect my ignorance of phylogenetic methods than anything else...). I think the main conclusions are well-supported by the data and the manuscript lays out the data and interpretations clearly. A few of the wider interpretations about the causes might be a stretch; for example lines 198-199: is there evidence for abnormal selective conditions, as opposed to perhaps broader environmental tolerances of newly-evolved clades? That’s somewhat of a minor issue that doesn’t detract from the overall claim of the manuscript, which I think will be of broad interest.

If I understand the method accurately, it seems that average clade age may constrain the possible range of pBC. If most branching points are deep back in time, pBC is constrained to be small (in the extreme, but presumably unrealistic, case where all branching points occurred earlier than the cutoff time μ , the value of pBC would be zero- I think?). Presumably this effect explains the slopes in figure S1. If this is the case, do your results simply measure clade age rather than ecological or biogeographic processes that promote cosmopolitan distributions? I expect that many new clades should radiate in the aftermath of extinctions. I suppose that there is still some biogeographic signal, as related taxa need to be found in separate regions (rather than the clade only radiating within a single region) to increase pBC, but do you have a sense for the relative magnitude of a biogeographic signal above and beyond just having younger clades?

On a related note, if clade age is important, do recently-evolved clades consistently exhibit greater pBC? Figure 4 shows that novel clades in the Early Triassic and early Early Jurassic have greater pBC, but would that be true in all time intervals? I don’t imagine that the absolute pBC value for novel clades would be as high in the Lopingian as it is in the Early Triassic, but I wonder if there would be some offset between continuous and novel clades in all time periods.”

Strictly speaking pBC could only be zero in the unrealistic edge case where there are both: 1) no shared taxa between localities, and 2) the phylogenetic hypothesis used represents a complete polytomy. In practice the latter almost never occurred in any data set (empirical or simulated) even after applying reasonable values for the cutoff, but the motivation for pBC was partially as a correction for the (much more realistic) former scenario (poorer fossil records tending towards a domination of singletons).

The cutoff, μ , was included in order to minimize the influence of clade age on the results, through maintaining a constant maximum branch length between bins and reducing the range in clade ages. However, we do acknowledge that by only presenting complete results we provided insufficient information to disentangle the influence that the relative proportion of young clades may have upon the result. To address this, we have performed new sensitivity analyses comparing the pBC of young clades through the Lopingian–Early Jurassic. The observed signal preserves the same relative (and in most bins, absolute) patterns through this interval to those observed in the original analysis. This indicates that abnormal biogeographic processes were indeed acting to generate high levels of pBC following extinction intervals, as opposed to being merely an artefact of average clade age in each interval.

“More generally, is the pBC value a non-unique solution? That is, could the same value arise from a situation with more links among moderately-related taxa and also from one with fewer links among more closely-related taxa? When thinking about a cosmopolitan disaster fauna, I don’t imagine most people would picture moderately-related taxa occurring in different regions. Would it be possible to parse the pBC value somehow to separate the contributions of taxa at different levels of relatedness? I suppose this is kind of a fuzzy question – maybe it wouldn’t provide any insights. That said, the values are less intuitive than the non-phylogenetic version (it’s easy to interpret when the exact taxon is shared), so it might be useful to help people gain that intuitive sense of the change in biogeographic connectedness (especially people like me who aren’t as well-versed in phylogenetic thinking).”

Although the scenario raised by the reviewer is theoretically possible we are not sure that it can be meaningfully disentangled. Some consideration of this point has been added in lines 330-337 of the main text, and in the supplementary information. Comparison of phylogenetic distances (see supplementary information) indicates that pBC results are not merely a result of sampling more proximate taxa in certain bins, but atomizing the signal further is difficult. A major problem with the proposed further analyses is how to examine taxa at equivalent taxonomic ranks: traditional Linnaean ranks like families have been abandoned by many vertebrate palaeontologists due to their subjectivity and lack of a consistent definition, and so cannot be applied across the entire supertree.

Consequently, considering only taxa of a certain phylogenetic “rank” is complicated by subjectivity, particularly so in the case of some Permo-Triassic tetrapods with complicated taxonomic histories. Something similar to this was attempted in analyses including only clades of a particular size across each of the extinction boundaries. However, these are highly vulnerable to differential sampling, one of the phenomena that this approach was purposefully intended to remedy. Consequently, we are not convinced that atomizing the signal across the entire interval in such a way would be meaningful, and the increased subjectivity would actually render the results less intuitive.

“A minor point, on lines 324-326: when the term “disaster taxa” was first coined, it referred to long-ranging organisms that were typically rare or restricted but that thrived in the extinction aftermath. These were things like stromatolites and Lingula (or really “Lingula”) after the Permian-Triassic extinction in the marine realm. The term is certainly used more broadly now, but I don’t know that I would have assumed recently-diverging clades to comprise the disaster fauna. Your results may support the idea that recently-diverging clades have less biogeographic differentiation (using that definition of a disaster fauna), assuming it’s not just a signal of average clade age.”

We based this definition of a “disaster fauna” on the concept that ‘weedy’ taxa surviving each mass extinction then diversified into widespread geographic ranges, leading to the lag between taxonomic and ecological recovery (e.g., Sahney & Benton, 2008; Chen & Benton, 2012). We acknowledge that this is divergent from the original descriptions of the term as described above: our meaning and reasoning behind our decision has been expanded upon in lines 357-365.

“Lines 150-152. At the risk of getting into semantics, is a pBC value of 0.05 (for the early Early Jurassic) really a global cosmopolitan fauna? If a value of 1 is truly cosmopolitan (albeit unrealistically so), 0.05 seems much less!”

This is true: our intention was to highlight the relative increase in pBC here. To put it another way we are highlighting peaks against a background trend of a decline, in much the same way as Raup and Sepkoski (1982; *Science*) did with mass versus background extinctions. This has now been clarified in the text.

“Finally, I’m not entirely convinced that there is a robust long-term trend towards more endemism. Yes, the Lopingian pBC value is higher than the Late Triassic or Late Early Jurassic, but the time-series is quite short. The Lopingian is also not long after a turnover in tetrapod faunas (Olson’s extinction). Without having a more data in the Paleozoic, I might be more circumspect about the significance of a background trend in endemism.

Sincerely,

Matthew Clapham”

This critic is valid and testing these patterns across longer time spans is a desirable avenue of continued study (and one of the key reasons we wish to share this method for broader use). However, an overall decline in pBC is still clear at least for the time series sampled by our study (i.e., the first 70 million years of the Mesozoic). This has now been clarified in lines 222-225.

References for responses

Benson, R.B.J. & Choiniere, J.N. Rates of dinosaur limb evolution provide evidence for exceptional radiation in Mesozoic birds. *Proc. R. Soc. B.* **280**, 20131780 (2013).

Button, D.J., Barrett, P.M. & Rayfield, E.J. 2Craniodental functional evolution in sauropodomorph dinosaurs. *Paleobiology* 1-28 (2017).

Kammerer, C. F., Fröbisch, J. & Angielczyk, K. D. On the validity and phylogenetic position of *Eubrachiosaurus browni*, a kannemeyeriiform dicynodont (Anomodontia) from Triassic North America. *PLoS One* **8**, e64203 (2013).

Ezcurra, M.D. The phylogenetic relationships of basal archosauromorphs, with an emphasis on the systematics of proterosuchian archosauriforms. *PeerJ* **4**, e1778 (2016).

Ezcurra, M.D., Montefeltro, F. & Butler, R.J. The early evolution of rhynchosaurs. *Front. Ecol. Evol.* **3**, 142 (2016).

Foth, C. & Joyce, W.G. Slow and steady: The evolution of cranial disparity in fossil and recent turtles. *Proc. R. Soc. B.* **283**, 20161881 (2016).

Foth, C., Ezcurra, M.D., Sookias, R.B., Brusatte, S.L. & Butler, R.J. Unappreciated diversification of stem archosaurs during the Middle Triassic predated the dominance of dinosaurs. *BMC Evolutionary Biology* **16**, 188 (2016).

Huttenlocker, A.K. Body Size Reductions in Nonmammalian Eutheriodont Therapsids (Synapsida) during the End-Permian Mass Extinction. *PLoS One* **9**, e87553 (2014).

Lloyd, G.T., Bapst, D.W., Friedman, M. & Davis, E.K. Probabilistic divergence time estimation without branch lengths: dating the origins of dinosaurs, avian flight and crown birds. *Biology Letters* **12**, 20160609 (2016).

Nesbitt, S.J. The early evolution of archosaurus: Relationships and the origins of major clades. *Bull. Am. Museum Nat. Hist.* **352**, 1-292 (2011).

Puttick, M.N., Thomas, G.H. & Benton, M.J. High rates of evolution preceded the evolution of birds. *Evolution* **68**, 1497-1510 (2014).

Stubbs, T.L. & Benton, M.J. Ecomorphological diversifications of Mesozoic marine reptiles: The roles of ecological opportunity and extinction. *Paleobiology* **42**, 547-573 (2016).

REVIEWERS' COMMENTS:

Reviewer #1 (Remarks to the Author):

I appreciate the responses given by the authors to the two main comments I had on the manuscript. The possibility that the southern pattern may arise from incomplete sampling is interesting and surely it will prompt further research in this issue.

Similarly, I am happy to see their expanded explanation in the response letter and the modified version of the manuscript.

I am glad to consider these clarifications sufficient regarding my previous concerns and I consider the manuscript should be accepted for publication.

Reviewer #2 (Remarks to the Author):

I think this revised version makes a very strong case for unique and predictable biogeographic changes in the aftermath of mass extinctions and for the broader macroevolutionary dynamics during extinction recoveries. The phylogenetic biogeographic methods are a nice methodological advance, especially when dealing with sparse and singleton-dominated vertebrate data.

I don't have any additional comments and I think the authors did an excellent job of addressing the reviewer comments.

Sincerely,

Matthew Clapham